# Deep Clustering with Uniform Quasi-low-rank Hypersphere Embedding

## Abstract

With the powerful representation ability of neural networks, deep clustering (DC) has been widely studied in machine learning communities. However, current research on DC has rarely laid emphasis on the inter-cluster representation structures, i.e. ignoring the performance degradation caused by the low uncorrelation between different clusters. To tackle this problem, a Uniform quasi-Low-rank Hypersphere Embedding based DC (ULHE-DC) method is proposed herein, which promotes learning an inter-cluster uniform and intra-cluster compact representation in a novel geometric manner. Specifically, clusters are uniformly distributed on a unit hypersphere via minimizing the hyperspherical energy of the centroids, and the embeddings belonging to the same cluster are simultaneously collapsed to a quasi-low-rank subspace through intra-cluster correlation maximization. Additionally, a pre-training based optimization scheme is proposed, in which an auto-encoder (AE) is pre-trained and the parameters of the encoder of AE are inherited to initialize the feature extractor for clustering, aiming at engaging the model learning cluster-oriented representation more efficiently. Experimental results validate the strong competitiveness of the proposed method, compared with several state-of-the-art (SOTA) benchmarks.

## 1 Introduction

Clustering is widely studied in numerous machine learning communities (Ehsan & René, 2013; Mathilde et al., 2018), such as computer vision, data mining, etc. As an unsupervised learning (Xu & Wunsch, 2005) based technology, clustering aims at learning a partition, ensuring similar samples belonging to the same cluster while grouping dissimilar ones into different clusters, and naturally possesses the technological advantage (i.e. annotation-free) compared with supervised learning. Conventional clustering methods, such as $k$-means (MacQueen, 1967), Gaussian mixture model (GMM) (Bishop, 2006), kernel $k$-means (Liu et al., 2016) and spectral clustering (SC) (Ng et al., 2001), group samples based on the intrinsically similar features or linear transformation of the raw data. However, these methods suffer from issues caused by the inflexibility of the hand-crafted feature or the incapacity to model the non-linear nature, and generally come under the performance degeneration and high computational complexity when dealing with high-dimensional and large-scale data.

Noting the superiority of deep neural networks (DNNs) on the ability of nonlinear representation, deep clustering (DC) methods have been proposed recently, which integrate deep learning to effectively learn more discriminative representation and capture the non-linear property. In general, the basic framework of DC typically comprises the auxiliary loss and clustering loss, respectively learning feasible features and inducing the cluster formation of feature embeddings. Specifically, the auxiliary loss can generally be the reconstruction loss (Dizaji et al., 2017; Lv et al., 2021), the variational loss (Jiang et al., 2017), or the adversarial loss (Mukherjee et al., 2019). The clustering loss can be the loss of any existing clustering algorithms, such as k-means, GMM, and hierarchical clustering. Nonetheless, DC needs to tackle with the following two optimization problems: **1) Intra-cluster compactness minimization.** Features of samples belonging to the same cluster should be highly correlated. **2) Inter-cluster discriminability maximization.** Samples belonging to different clusters should be embedded in the feature space with extremely low correlation.

However, most existing DC approaches mainly focus on the first issue and learn suitable embeddings with the DNNs trained through a clustering-oriented loss function, which causes that hard samples near the cluster boundaries cannot supply enough representation guidance. In addition, few researches on DC explicitly pay attention to the second problem. Coincidentally, recent studies (Hu et al., 2014; De et al., 2016) on the supervised tasks have similar properties, which performed the minimization of the Euclidean distance between the deep intra-class embeddings but keeping the inter-class ones apart. More recently, an orthogonal low-rank embedding (OLE) (Lezama et al., 2018) loss was proposed to encourage the neural networks to learn more discriminative features, subspaces of which are intra-class low-rank regularized but inter-class orthogonalized at the same time. The OLE promotes the network to learn one-dim representations for each category but limits the utilization of the whole space, compared with the uniform embeddings of cluster centroids. Besides, the nuclear norm in the OLE loss function is non-smooth, which potentially raises difficulties during the gradient descent based optimization. To alleviate these problems, a representation learning framework based on maximal coding rate reduction (Yu et al., 2020) was proposed to learn subspaces with maximal dimensions, trained with a determinant based smooth loss. Whereas, the determinant operator will result in the computational complexity explosion when the batch size is relatively large.

Addressing the above issues, a Uniform quasi-Low-rank Hypersphere Embedding based DC (ULHE-DC) method is proposed in this paper, including pretraining and clustering two stages. Firstly, an autoencoder is trained by minimizing the reconstruction and normalizing each embedding on the unit hypersphere, transforming data to low dimensional representation space. Then, the encoder is finetuned by using the basic clustering loss. Additionally, the ULHE is designed as a regularizer for the clustering loss, composed of the minimization of the hyperspherical energy between cluster centroids and the maximization of the correlation between members of each cluster, which respectively stimulate the learning preference of the model to uniformly embed the cluster centroids on hypersphere, enhancing the inter-cluster discriminability and diversity, and generate quasi-low-rank and compact embeddings of members belonging to the same cluster. In particular, the formulation of ULHE based loss is smooth and computationally friendly. Main contributions can be summarized as follows:

- A novel framework named Uniform quasi-Low-rank Hypersphere Embedding based DC (ULHE-DC) is proposed to optimize the cluster-oriented presentation structure, which can be efficiently implemented with a mini-batch based learning strategy.

- ULHE is established to enhance the inter-cluster discriminability and diversity with minimizing the hyperspherical energy, encouraging the centroids being uniformly embedded on the hypersphere; meanwhile, it enforces the feature embeddings of the same cluster squashed in a quasi-low-rank subspace through the maximization of intra-cluster correlation.

- Extensive experiments validate the effectiveness and superiority of ULHE-DC via comparing with several state-of-the-art (SOTA) DC approaches on four benchmarks.

## 2 RELATED WORK

**Deep Clustering.** DC is a family of clustering algorithms that adopt DNNs to learn cluster-oriented representations. From the perspective of the type of DNNs, DC approaches can be divided into four categories: AE-based, Variational autoencoder (VAE) (Kingma & Welling, 2013) based, generative adversarial network (GAN) (Goodfellow et al., 2014) based, and clustering DNN (CDNN) based. As an extensively studied branch of DC, AE-based DC integrates prior knowledge into the objective function of AE. The clustering loss functions are mainly the objective of $k$-means (Yang et al., 2017; Fard et al., 2020), the variant objective of $k$-means (Jabi et al., 2021), or the other kinds of loss (Ji et al., 2017). The superiority of AE-based DC is that the scheme of conventional clustering and the regularization of feature embedding can be reasonably employed to the training procedure of AE. VAE-based DC (Jiang et al., 2017; Dilokthanakul et al., 2016) prefers learning a representation, which follows a predefined distribution of the cluster structure, but suffers from high computational complexity. GAN-based methods (Chen et al., 2016; Zhou et al., 2018; Mukherjee et al., 2019) enforce the embedding of the deep feature in a similar way as VAE-based ones. However, the model collapse problem and the training challenge of GAN also exist. CDNN-based algorithms (Peng

et al., 2017; Guérin & Boots, 2018; Deng et al., 2023) train the extractor merely with the clustering loss, which may result in obtaining corrupted feature space, that is, a convergent loss possibly makes no sense. Recently, those existing DC approaches have been proposed mainly from the perspective of network architectures, various clustering loss or tricks in deep learning. The proposed ULHE regularizer is introduced to restrain the latent embeddings, which keeps intra-cluster members compact and inter-cluster ones relatively uniform on a unit hypersphere.

**Minimum Hyperspherical Energy (MHE).** Drawing inspiration from the Thomson problem in physics, MHE (Liu et al., 2018) is defined to seek the equilibrium state of the distribution of mutually repelling electrons through minimizing the potential energy. More generally, lower energy indicates more diverse and more uniform distribution. MHE has been extensively researched, which shows noteworthy effectiveness in many applications. MHE was firstly proposed and used as a generic regularization for neural networks in (Liu et al., 2018), regularizing the networks to avoid representation redundancy. Analogously, the compressive minimum hyperspherical energy (Lin et al., 2020) and the hyperspherical uniformity regularization (Liu et al., 2021) were established. A MHE-based active learning algorithm (Cao et al., 2023) was designed to effectively characterize the decision boundaries for data learning. Besides, MHE has been widely applied in image recognition (Chen et al., 2020; Li et al., 2020), speaker verification, adversarial robustness (Pang et al., 2019), etc. In DC, maximizing the inter-cluster discriminability is approximated to enhance the diversity of clusters, which can be implemented through embedding the centroids as evenly as possible, and MHE provides a solution from a geometric perspective.

# 3 METHODOLOGY

## 3.1 FRAMEWORK OVERVIEW

Given an unlabeled dataset $\mathcal{X} = \{\boldsymbol{x}_i \in \mathbb{R}^D\}_{i=1}^N$, deep clustering aims to assign $N$ samples to $K$ clusters. Note that $K$ is priorly given in this study. In deep clustering, samples are generally mapped to a much lower dimension feature space with an embedding network $F_{\boldsymbol{w}} := \boldsymbol{x}_i \to \boldsymbol{z}_i, \boldsymbol{z}_i \in \mathbb{R}^d (d \ll D)$. With parameters $\boldsymbol{w}$ well optimized by minimizing the clustering loss function, the embedding network is expected to extract more suitable feature for clustering.

The proposed ULHE-DC method aims to learn cluster-oriented features based on an AE networks and includes the pretraining and clustering two stages. Firstly, the AE is pretrained to extract feasible features with the reconstruction loss $\mathcal{L}_{rec}$ and a normalized loss $\mathcal{L}_{norm}$ to embed data on a unit hypersphere. After pretraining, ULHE-DC finetunes the encoder part of AE both with the clustering objective and the ULHE based regularization loss $\mathcal{L}_{unif}$ and $\mathcal{L}_{cmpt}$, making the learned representations cluster-friendly.

## 3.2 BASIC DEEP CLUSTERING MODEL

**Pretraining Stage.** The AE, composed of the encoder network $F_{\boldsymbol{w}}(\cdot)$ and the decoder network $G_\theta(\cdot)$, is trained towards minimizing the sum of $\mathcal{L}_{rec}$ and $\mathcal{L}_{norm}$, which are respectively formulated as

$$\mathcal{L}_{rec} = \mathbb{E}_{\boldsymbol{x}_i \sim \mathcal{X}} \|\boldsymbol{x}_i - G_\theta(F_{\boldsymbol{w}}(\boldsymbol{x}_i))\|_2^2 \tag{1}$$

and

$$\mathcal{L}_{norm} = \mathbb{E}_{\boldsymbol{x}_i \sim \mathcal{X}} (\|F_{\boldsymbol{w}}(\boldsymbol{x}_i)\|_2 - 1)^2, \tag{2}$$

where $\|\cdot\|_2$ denotes the $l_2$-norm projection, and the whole pretraining loss is

$$\mathcal{L}_{norm-rec} = \mathcal{L}_{rec} + \mathcal{L}_{norm}. \tag{3}$$

In pretraining, the encoder $F_{\boldsymbol{w}}(\cdot)$, serves as a powerful feature extractor to transform the data $\boldsymbol{x}_i$ to a low dimensional embedding $\boldsymbol{z}_i$. As the pretraining scheme is not task-oriented, hence $\boldsymbol{z}_i$ is not suitable for clustering and $F_{\boldsymbol{w}}(\cdot)$ needs to be finetuned with a clustering loss.

**Clustering Stage.** On account of the representation embedded on hypersphere, the clustering objective is similar to that of $k$-means, in which the Euclidean distance is replaced by the cosine distance,

and can be defined as

$$\min_{\boldsymbol{w},\boldsymbol{M},\boldsymbol{s}} \mathbb{E}_{\boldsymbol{x}_i \sim \mathcal{X}} 1 - cos(F_{\boldsymbol{w}}(\boldsymbol{x}_i), \boldsymbol{M}\boldsymbol{s}_i), s.t. \boldsymbol{s}_i \in \{0,1\}^K, \mathbf{1}^\mathsf{T}\boldsymbol{s}_i = 1. \tag{4}$$

where $\boldsymbol{M} = \{\boldsymbol{m}_k | \boldsymbol{m}_k \in \mathbb{R}^{d \times 1}\}_{k=1}^K \in \mathbb{R}^{d \times K}$ denotes the centroid matrix, i.e. each column corresponding to a cluster center, $\boldsymbol{s}_i \in \mathbb{R}^{K \times 1}$ is the assignment of $\boldsymbol{x}_i$ and $\mathbf{1}$ is a column vector with all the elements set to 1. First and foremost, the centroid matrix $\boldsymbol{M}$ is initialized with a variant of $k$-means, the objective of which can be rewritten as

$$\min_{\boldsymbol{M},\boldsymbol{s}} \mathbb{E}_{\boldsymbol{x}_i \sim \mathcal{X}} 1 - cos(F_{\boldsymbol{w}}(\boldsymbol{x}_i), \boldsymbol{M}\boldsymbol{s}_i), s.t. \boldsymbol{s}_i \in \{0,1\}^K, \mathbf{1}^\mathsf{T}\boldsymbol{s}_i = 1. \tag{5}$$

It performs clustering through alternatively updating the assignments $\boldsymbol{s}$ and cluster centroids $\boldsymbol{M}$ respectively with

$$s_{j,i} = \begin{cases} 1, if\ j = \underset{k=\{1,2,...,K\}}{argmin}\ 1 - cos(F_{\boldsymbol{w}}(\boldsymbol{x}_i), \boldsymbol{m}_k) \\ 0, otherwise, \end{cases} \tag{6}$$

where $s_{j,i}$ is the $j$-th element of $\boldsymbol{s}_i$, $\boldsymbol{m}_k$ is the $k-th$ cluster centroid, and

$$\boldsymbol{m}_k = Norm\left(\sum_{i \in \mathcal{C}_k} \frac{F_{\boldsymbol{w}}(\boldsymbol{x}_i)}{\|F_{\boldsymbol{w}}(\boldsymbol{x}_i)\|}\right), \tag{7}$$

where $\mathcal{C}_k$ is the index set of samples assigned to the $k$-th cluster and $Norm(\cdot)$ is the function to normalize the norm of a vector to 1. Nevertheless, the above updation of cluster centroids is problematic, since the current samples in the mini-batch is not enough to model the global cluster structure and the assignments might change. To alleviate this problem, the $k$-th centroid $\boldsymbol{m}_k^{(t)}$ in the $t$-th iteration is updated by the weighted $\boldsymbol{m}_k^{(t-1)}$ and the temporary centroid $\hat{\boldsymbol{m}}_k^{(t)}$ of newly assigned samples as follows:

$$\boldsymbol{m}_k^{(t)} = Norm\left(\boldsymbol{m}_k^{(t-1)} + \frac{K|\mathcal{C}_k^{(t)}|}{N}\hat{\boldsymbol{m}}_k^{(t)}\right), \tag{8}$$

where $|\mathcal{C}_k^{(t)}|$ is denoted as the number of samples assigned to the $k$-th cluster in the $t$-th iteration and $\hat{\boldsymbol{m}}_k^{(t)}$ can be calculated by Eq. (7).

With the basic deep clustering model, it implements clustering via alternatively optimizing Eq. (4) to learn cluster-oriented representation and updating the assignments $\boldsymbol{s}$ the centroids matrix $\boldsymbol{M}$ respectively by Eq. (6) and Eq. (8). In contrast to the supervised learning, it can not guarantee that samples currently assigned to the same cluster remain in the same one during the whole clustering stage. Therefore, it makes restricted contribution to learning discriminative and diverse inter-cluster representation structures merely relying on optimizing the basic clustering objective in Eq. (4). To accomplish this aim, a ULHE based regularization loss is added to the above objective.

## 3.3 UNIFORM QUASI-LOW-RANK HYPERSPHERICAL EMBEDDING

Towards learning a more cluster-friendly representation, the ULHE regularizer is incorporated to the clustering objective mentioned above, which indeed includes an inter-cluster uniformity loss, enhancing the centroids uniformly embedded within the representation space, and an intra-cluster compactness loss, enforcing a quasi-low-rank constraint on features of the same cluster.

**Inter-cluster Uniformity Regularization.** Aiming at ensuring the discriminability and diversity between clusters, it is intuitive that all the clusters are expected to be uniformly distributed in the representation space. Inspired by the well-known Thomson problem, the goal can be accomplished with the minimization of the potential energy of all the centroids. Given $K$ cluster centroids, i.e. $\boldsymbol{M} = [\boldsymbol{m}_1, \boldsymbol{m}_2, ..., \boldsymbol{m}_K]^\mathsf{T}$, then their hyperspherical energy can be formulated as

$$\mathcal{E}_v(\boldsymbol{m}_k|_{k=1}^K) := \sum_{i=1}^K \sum_{j=1}^K f_v(\|\boldsymbol{m}_i - \boldsymbol{m}_j\|)$$

$$= \begin{cases} \sum_{i>j} \|\boldsymbol{m}_i - \boldsymbol{m}_j\|^{-v}, & v > 0 \\ \sum_{i>j} log(\|\boldsymbol{m}_i - \boldsymbol{m}_j\|^{-1}), & v = 0 \end{cases}, \tag{9}$$

where $f_v(\cdot)$ is an energy function. It is obvious that the argument of the hyperspherical energy function $\mathcal{E}_v$ only contains the parameter of the encoder network, namely, $\boldsymbol{w}$. Hence, the minimization problem is equivalent to optimizing $\boldsymbol{w}$. In order to simplifying the problem, $v$ is set to 2. Then, the optimization is defined as

$$\underset{\boldsymbol{w}}{argmin}\,\mathcal{E}_2(\boldsymbol{m}_k|_{k=1}^K) = \sum_{i>j} \|\boldsymbol{m}_i - \boldsymbol{m}_j\|^{-2}, \tag{10}$$

which can be simplified to

$$\begin{aligned}
\underset{\boldsymbol{w}}{argmin}\,\mathcal{E}_2(\boldsymbol{m}_k|_{k=1}^K) &= \sum_{i>j} \|\boldsymbol{m}_i - \boldsymbol{m}_j\|^{-2} \\
&= \sum_{i>j} 1/\left(\|\boldsymbol{m}_i\|^2 + \|\boldsymbol{m}_j\|^2 - 2\boldsymbol{m}_i^\mathsf{T}\boldsymbol{m}_j\right) \\
&= \sum_{i>j} 1/\left[2(1 - \boldsymbol{m}_i^\mathsf{T}\boldsymbol{m}_j)\right],
\end{aligned}$$

due to $\|\boldsymbol{m}_k\| = 1$, for $k = 1, 2, ..., K$. More specifically, as a result of

$$sum(1/\left[2(\boldsymbol{1} - \boldsymbol{M}^\mathsf{T}\boldsymbol{M}\right]) = \sum_{i>j, i=j, i<j} 1/\left[2(1 - \boldsymbol{m}_i^\mathsf{T}\boldsymbol{m}_j)\right],$$

where $sum(\cdot)$ is a function to calculate the sum of all the elements of a matrix,

$$\sum_{i>j} 1/\left[2(1 - \boldsymbol{m}_i^\mathsf{T}\boldsymbol{m}_j)\right] = \sum_{i<j} 1/\left[2(1 - \boldsymbol{m}_i^\mathsf{T}\boldsymbol{m}_j)\right]$$

and $\sum_{i=j} 1/\left[2(1 - \boldsymbol{m}_i^\mathsf{T}\boldsymbol{m}_j)\right] = 0$, the inter-cluster uniformity regularization loss can be formulated as

$$\mathcal{L}_{unif}(\boldsymbol{w}) = sum(1/(\boldsymbol{1} - \boldsymbol{M}^\mathsf{T}\boldsymbol{M})), \tag{11}$$

according to Eq. (10). Note that the centroid matrix $\boldsymbol{M}$ in $\mathcal{L}_{unif}$ is computed with samples in the current mini-batch to accommodate the batch optimization.

**Intra-cluster Compactness Regularization.** Considering that $\mathcal{L}_{unif}$ is calculated with centroids in the mini-batch, it may be unstable while the intra-cluster embeddings are not enough compact. Consequently, it is of great necessity that the learned intra-cluster features should be highly correlated and coherent, i.e. each cluster should only span a low-rank subspace. which is equivalent to maximizing the intra-cluster hyperspherical energy. Or rather, the total linear correlation (or similarity) of feature vectors between each other should be as high as possible. Let $\boldsymbol{Z}_k = \{F_{\boldsymbol{w}}(\boldsymbol{x}_i)|i \in \mathcal{C}_k\} \in \mathbb{R}^{d \times |\mathcal{C}_k^i|}$ denote the embedding matrix of data in the $k$-th cluster, and it is readily comprehensible that the larger intra-cluster energy is, the more compact feature embeddings $\boldsymbol{z}_i$ are, as opposed to MHE. Moreover, maximization of the intra-cluster energy means that the cosine similarity of features in the same cluster should be at a high level, which can be formulated as

$$\mathcal{L}_{cmpt}(\boldsymbol{w}) = \frac{1}{K}\sum_{k=1}^K sum(\boldsymbol{1} - \boldsymbol{Z}_k^\mathsf{T}\boldsymbol{Z}_k). \tag{12}$$

Next, a brief proof is given to indicate that minimizing $\mathcal{L}_{cmpt}$ provides a guidance for $F_{\boldsymbol{w}}(\cdot)$ to learn a quasi-low-rank structure in the intra-cluster representations. According to the *Eckart-Young Theorem*, suppose $\boldsymbol{Z}_k = \boldsymbol{A} = \boldsymbol{U}\boldsymbol{\Sigma}\boldsymbol{V}^\mathsf{T}$ is the singular value decomposition (SVD) of intra-cluster embeddings $\boldsymbol{Z}_k$, with singular values $\sigma_1 \geq \sigma_2 \geq ... \geq \sigma_p \geq 0$. Let $r < R = rank(\boldsymbol{A})$ and the truncated matrix $\boldsymbol{A}_r = \sum_{i=1}^r \sigma_i \boldsymbol{u}_i \boldsymbol{v}_i^\mathsf{T}$, then for any matrix B of rank $r$, the minimal error of Frobenius norm is achieved with $\boldsymbol{A}_r$:

$$\min_{rank(\boldsymbol{B})=r} \|\boldsymbol{A} - \boldsymbol{B}\|_F^2 = \|\boldsymbol{A} - \boldsymbol{A}_r\|_F^2 = \sum_{i=r+1}^p \sigma_i^2,$$

where $\|\cdot\|_F$ denotes the Frobenius norm projection. That is, $\boldsymbol{A}_r$ is the best low-rank approximation of $\boldsymbol{A}$ and the error $\|\boldsymbol{A}_r - \boldsymbol{B}\|_F^2$ can be further minimized through the optimization of $\boldsymbol{w}$. In the case of the limit situation, suppose $r = 1$, it indicates that $rank(\boldsymbol{Z}_k) \approx 1$ if $\sum_{i=2}^p \sigma_i^2$ has been minimized to a small value, which means that the embeddings $\{F_{\boldsymbol{w}}(\boldsymbol{x}_i)|i \in \mathcal{C}_k\}$ maintain a relatively small cosine distance between each other. Due to $\|F_{\boldsymbol{w}}(\boldsymbol{x}_i)\|_2 = 1$, the formulation $\frac{1}{2}sum(\boldsymbol{1} - \boldsymbol{Z}_k^\mathsf{T}\boldsymbol{Z}_k)$ is indeed the total cosine distance of samples in the $k$-th cluster. Therefore, minimizing the intra-cluster compactness regularization loss Eq. (12) will squash the examples in the same cluster to a quasi-low-rank subspace, compared with the OLE. Besides, it avoids the extremely complex computation of singular value of $\boldsymbol{Z}_k$.

## 3.4 OPTIMIZATION

The training procedure can be clearly compartmentalized to two stage, i.e. the pretraining and clustering stage. In the following, the optimization strategy and stopping criterion are introduced. Furthermore, the computational complexity is analyzed.

**Optimization Strategy.** In the pretraining stage, the encoder $F_w(\cdot)$ can be directly optimized by the SGD optimizer and backpropagation. During clustering, the assignments $s$ and the centroid matrix $M$ are respectively updated with Eq. (6) and Eq. (8) when $w$ fixed. Then with $s$ and $M$ fixed, $w$ is updated by minimizing the weighted objective

$$\min_{w} \mathbb{E}_{x_i \sim \mathcal{X}}[1 - cos(F_w(x_i), Ms_i)] + \lambda_0 \mathcal{L}_{norm}$$
$$+ \lambda_1 \mathcal{L}_{unif} + \lambda_2 \mathcal{L}_{cmpt}, s.t. s_i \in \{0,1\}^K, \mathbf{1}^\mathsf{T} s_i = 1, \tag{13}$$

where $\lambda_0$, $\lambda_1$ and $\lambda_2$ are weights to balance the basic clustering objective (4), the normalized loss $\mathcal{L}_{norm}$, the inter-cluster uniformity regularization loss $\mathcal{L}_{unif}$ and the intra-cluster compactness regularization loss $\mathcal{L}_{cmpt}$.

**Stopping Criterion.** For the sake of obtaining a stable but not degenerated model, the clustering training will stop, if the change rate of cluster assignments between two successive iterations is lower than a manually set threshold $\eta$. Then, the stopping criterion is defined as

$$1 - sum(s^{(t-1)} \odot s^{(t)})/N < \eta, \tag{14}$$

where $\odot$ is signfied as an element-wise multiplication operator for two matrices.

**Computational Complexity.** Finally, the computational complexity of the proposed ULHE-DC is analyzed. Suppose $\tilde{N}$ denotes the maximum number of neurons in each layer of the AE and the pretraining epochs is $T_1$, then the time complexity of pretraining AE is $\mathcal{O}(T_1 \tilde{N}^2 N)$. For the clustering stage, the time complexity of the initialization of $M$ and $s$ is $\mathcal{O}(T_2 K dN)$, where $T_2$ is the iterations of the mentioned variant of $k$-means, and those of updating $s$ and $M$ are $\mathcal{O}(TKdN)$ and $\mathcal{O}(TdN)$, respectively. Via minimizing Eq. (13), $w$ is updated with a relatively high computational complexity $\mathcal{O}(T\tilde{N}^2 dN^2/K)$, due to the matrix multiplication in the ULHE based regularization loss. The total time complexity of ULHE-DC is $\mathcal{O}(T_1 \tilde{N}^2 N + (T_2 + T)KdN + T\tilde{N}^2 dN^2/K)$. Though the total time complexity is not linear to the number of samples $N$, the efficiency can be improved through the mini-batch optimization.

## 4 EXPERIMENTS

### 4.1 DATASETS AND METRICS

**Benchmark Datasets.** To validate the proposed method performing well on various datasets, four image datasets are chosen to conduct the experiments , details of which are described as follows. The first dataset is MNIST-*full* (Yann et al., 1998), which totally consists of 70,000 handwritten digits, including 10 categories and each monochrome image with the size of $28 \times 28$. The second one is MNIST-*test*, which only contains the testing set of MNIST-*full*, namely 10,000 samples. USPS is selected as the third, composed of 9298 $16 \times 16$ handwritten digit images in total and divided into 10 classes, which is then split into 7291 training images and 2007 test images. The last one is Fashion (Han et al., 2017), which is more complicated and collects 70,000 $28 \times 28$ gray images, including 10 categories of articles on Zalando.

**Evaluation Metrics.** The clustering ACCuracy (ACC) and Normalized Mutual Information (NMI) are applied as standard metrics to evaluate clustering approaches. The metric of ACC is defined as the best mapping between ground truth $\mathbf{y}$ and cluster assignments $\hat{\mathbf{y}}$, which can be formulated as

$$ACC = \max_{m} \frac{\sum_{i=1}^{N} \mathbf{1}(y_i = m(\hat{y_i}))}{N}, \tag{15}$$

where $y_i$ and $\hat{y_i}$ are respectively the true label and the cluster assignment of sample $x_i$, and $m$ is an over all one to one mappings between true labels and cluster assignments. which can be efficiently calculated by the Hungarian algorithm (Kuhn, 2005). The metric of NMI, measuring the normalized

Table 1: Comparison of clustering performances on four datasets. The best value and the second best vale are respectively highlighted in bold and underlined. The result of ULHE-DC is acquired with $\lambda_0 = 2.00$, $\lambda_1 = 0.08$ and $\lambda_2 = 0.40$.

| Methods | MNIST-*full* | | MNIST-*test* | | USPS | | Fashion | |
|---|---|---|---|---|---|---|---|---|
| | ACC | NMI | ACC | NMI | ACC | NMI | ACC | NMI |
| $k$-means | 0.5381 | 0.5047 | 0.5446 | 0.5013 | 0.6754 | 0.6307 | 0.4720 | 0.5114 |
| GMM | 0.4270 | 0.3563 | 0.5142 | 0.4815 | 0.5631 | 0.5373 | 0.5692 | 0.5615 |
| SC | 0.6560 | 0.7310 | 0.6600 | 0.7040 | 0.6490 | 0.7940 | 0.5080 | 0.5750 |
| DEC | 0.8630 | 0.8340 | 0.8560 | 0.8300 | 0.7620 | 0.7670 | 0.5180 | 0.5460 |
| JULE* | 0.9640 | 0.9130 | 0.9610 | 0.9150 | 0.9500 | 0.9130 | 0.5630 | 0.6080 |
| DEPICT* | 0.9650 | 0.9170 | 0.9630 | 0.9150 | 0.9241 | 0.9098 | 0.4406 | 0.4213 |
| ClusterGAN | 0.9500 | 0.8900 | – | – | – | – | 0.6300 | 0.6400 |
| VaDE | 0.9389 | 0.8734 | – | – | 0.5660 | 0.5120 | 0.5780 | 0.630 |
| DAC* | 0.9780 | 0.9350 | – | – | – | – | – | – |
| DSC-DAN* | 0.9780 | 0.9410 | 0.9800 | 0.9460 | 0.8690 | 0.8570 | 0.6620 | 0.6450 |
| DDC-DA* | 0.9690 | 0.9410 | 0.9700 | 0.9270 | 0.9770 | **0.9390** | 0.6090 | 0.6610 |
| SENet* | 0.9680 | 0.9180 | – | – | – | – | **0.6970** | 0.6630 |
| DeepDPM | 0.9793 | 0.9381 | – | – | 0.8950 | 0.8817 | 0.6242 | 0.6772 |
| DCCF* | 0.9741 | 0.9332 | – | – | 0.8553 | 0.8251 | 0.6212 | 0.6458 |
| DML-DSL* | 0.9636 | 0.9124 | – | – | – | – | 0.6320 | 0.6480 |
| ULHE-DC | **0.9836** | **0.9613** | **0.9812** | **0.9485** | **0.9788** | 0.9371 | 0.6440 | **0.6739** |
| | ±**0.0015** | ±**0.0023** | ±**0.0027** | ±**0.0014** | ±**0.0019** | ±0.0030 | ±**0.0125** | ±**0.0287** |

similarity between the ground truth and the cluster assignment of the same sample, is defined as

$$NMI = \frac{I(\mathbf{y}, \hat{\mathbf{y}})}{\max\{H(\mathbf{y}), H(\hat{\mathbf{y}})\}}, \tag{16}$$

where $I(\cdot)$ and $H(\cdot)$ denotes the mutual information and entropy, respectively. Both of the two metrics are normalized to the range of $[0, 1]$. Note that the higher the metrics are, the better the clustering performance is.

## 4.2 EXPERIMENTAL SETTING

About the network structure, ULHE-DC includes seven hidden fully connected layers with dimensions 500, 500, 2000, 10, 2000, 500, 500 respectively, the input and output dimensions of which are those of the input samples. In addition, all the hidden layers are activated by the rectified linear unit (ReLU) (Glorot et al., 2011).The experiments are all implemented with the PyTorch 2.0 framework on a single NVIDIA GeForce RTX 4090 with 24-GB RAM. In the pretraining stage, the AE is end-to-end trained wirh the SGD optimizer, the momentum of which was set to 0.90, and the batch size, the learning rate and training epochs are respectively set to 256, 0.10 and 1000. During clustering, the optimizer and batch size is with the same setting as above, while the learning rate and training iterations are changed to 0.002 and 300. Besides, hyperparameters $\lambda_0$, $\lambda_1$ and $\lambda_2$ are respectively set to 2.00, 0.08 and 0.40 to balance the components of Eq. (13). The threshold $\eta$ in Eq. (14) were set to 0.001. To stable the process of clustering, a simple self-paced learning (Kumar et al., 2010) schedule was introduced, in which samples were orderly fed into the model in three batches from easy to hard and the sample weights were updated every 100 epochs. More specifically, the closer the sample is to the cluster center, the easier it is. To obtain stable experiment results of the proposed method, all experiments were carried out five times on each dataset.

## 4.3 PERFORMANCE COMPARISON

The clustering performance of the proposed method, ULHE-DC, is compared with several baseline and SOTA DC approaches, which include $k$-means (MacQueen, 1967), GMM (Bishop, 2006), SC (Ng et al., 2001), deep embedded clustering (DEC) (Xie et al., 2016), joint unsupervised learning (JULE) (Yang et al., 2016), deep embedded regularized clusTering (DEPICT) (Dizaji et al.,

Table 2: Clustering performance with different regularization loss functions on MNIST-*full*.

| Model | ULHE Loss | | Metrics | |
|:---:|:---:|:---:|:---:|:---:|
| | $\mathcal{L}_{unif}$ | $\mathcal{L}_{cmpt}$ | ACC | NMI |
| 1 | – | – | $0.9186 \pm 0.0020$ | $0.8747 \pm 0.0035$ |
| 2 | ✓ | – | $0.9372 \pm 0.0041$ | $0.9011 \pm 0.0089$ |
| 3 | – | ✓ | $0.9665 \pm 0.0012$ | $0.9329 \pm 0.0018$ |
| 4 | ✓ | ✓ | $\mathbf{0.9836 \pm 0.0015}$ | $\mathbf{0.9613 \pm 0.0023}$ |

2017), clustering with GAN (ClusterGAN) (Mukherjee et al., 2019), variational deep embedding (VaDE) (Jiang et al., 2017), deep adaptive clustering (DAC) (Chang et al., 2017), dual AE based deep spectral clustering (DSC-DAN) (Yang et al., 2019), deep density-based clustering (DDC-DA) (Ren et al., 2020), SC with self-expressive network (SENet) (Zhang et al., 2021), deep nonparametric clustering method (DeepDPM) (Ronen et al., 2022), contractive feature representation based DC (DCCF) (Cai et al., 2022) and deep Multirepresentation Learning (DML-DSL) (Sadeghi & Armanfard, 2023). The clustering results of all methods are reported in Table 1. As far as the baseline algorithms are concerned, all the reported results were acquired through running the released code except the ones of methods marked by (*) on top, which are excerpted from the corresponding paper. Results marked by "–" denotes that they are unavailable from the paper.

As shown in Table 1, DC approaches, from DEC to ULHE-DC, outperform the conventional ones ($k$-means, GMM and SC) by a large margin in most situations, benefiting from the superior ability of feature extraction. Moreover, even on the most difficult dataset Fashion, ULHE-DC exceeds the best of shallow clustering methods GMM by $7.48\%$ and $11.24\%$, respectively on ACC and NMI. Compared with other DC methods, it can be noticed that ULHE-DC achieves the best performance in terms of ACC or NMI on all the four datasets, except NMI on USPS and ACC on Fashion. Especially when performing on the dataset MNIST-*full* and MNIST-*test*, the SOTA accuracies are both increased to $98.00\%$. In particular, on the most widely used MNIST-*full*, it exceeds the second best DeepDPM performance by $0.43\%$ and $2.03\%$ on ACC and NMI, respectively. Even with regard to the Fashion, which is the most difficult among the four datasets, ACC of ULHE-dc is not the best whereas comparable, but what is more remarkable is that ULHE-DC exceeds SENet by a margin of $1.09\%$ on NMI. Considering the different inter-class discriminability, hard samples can be more easily assigned with incorrect but the same label, because of the implement of *Intra-cluster Compactness Regularization*. That is, the distribution of $\{p(\mathbf{y}|\hat{\mathbf{y}}; \mathbf{y} \neq \hat{\mathbf{y}})\}_{K-1}$ is unbalanced, so the conditional entropy $H(\mathbf{y}|\hat{\mathbf{y}})$ is relatively small. Moreover, NMI can be written as $NMI = 2I(\mathbf{y}, \hat{\mathbf{y}})/(H(\mathbf{y})+H(\hat{\mathbf{y}})) = 2(H(\mathbf{y})-H(\mathbf{y}|\hat{\mathbf{y}}))/(H(\mathbf{y})+H(\hat{\mathbf{y}}))$. Due to the fact that datasets in the experiments are balanced, the denominators of NMIs on different methods are approximate while the ACCs close to each other. Hence, the enhancement in NMI is more noteworthy.

## 4.4 ABLATION STUDY

Two key components exist in the proposed ULHE-DC, the inter-cluster uniformity regularization loss $\mathcal{L}_{unif}$ and the intra-cluster compactness regularization loss $\mathcal{L}_{cmpt}$. To analyze the contribution of the components, the ablation study is conducted on MNIST-*full*. As shown in Table 2, different strategies of training models are: 1) Mode-1, the pretrained $F_{\mathbf{w}}(\cdot)$ with the clustering objective $\mathcal{L}_{clus}$ (Eq. (4)), named $F_{\mathbf{w}}(\cdot) + \mathcal{L}_{clus}$; 2) Mode-2 *w/o* $\mathcal{L}_{cmpt}$, ULHE-DC trained only without $\mathcal{L}_{cmpt}$; 3) Mode-3 *w/o* $\mathcal{L}_{unif}$, ULHE-DC trained only without $\mathcal{L}_{unif}$; 4) Mode-4, ULHE-DC trained by the clustering objective Eq.(13). Table 2 represents the performance of different strategies for training our model, with $\lambda_0$, $\lambda_1$ and $\lambda_2$ respectively set to 2.00, 0.08 and 0.40.

Some conclusions can be observed from Table 2. Above all, applying the inter-cluster uniformity regularization via adding $\mathcal{L}_{unif}$ to Model-1 and Model-3 could consistently improve the performance with the increase of $1.86\%$ and $1.71\%$ on ACC, respectively. It is mainly because that minimizing $\mathcal{L}_{unif}$ could assist the model to learn more discriminative and diverse inter-cluster representations. However, $\mathcal{L}_{unif}$ is relatively sensitive while the members of the same cluster are dispersed in a subspace, which results in the degradation of performance stability. Secondly, the intra-cluster compactness regularization makes more contribution for the clustering performance. Comparing

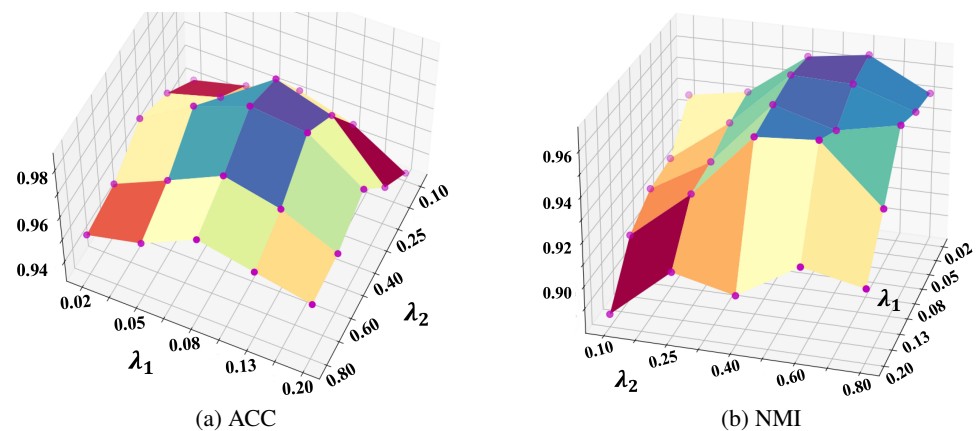

(a) ACC                    (b) NMI

Figure 1: ACC and NMI of ULHE-DC with different $\lambda_1$ and $\lambda_2$ on MNIST-*full*.

Table 3: Clustering performance with different hyperparameter settings on MNIST-*full*.

| Res. \ $\lambda_1$ | 0.02 | | 0.05 | | 0.08 | | 0.13 | | 0.20 | |
|---|---|---|---|---|---|---|---|---|---|---|
| $\lambda_2$ | ACC | NMI | ACC | NMI | ACC | NMI | ACC | NMI | ACC | NMI |
| 0.10 | 0.9422 | 0.9122 | 0.9543 | 0.9343 | 0.9615 | 0.9589 | 0.9519 | 0.9531 | 0.9490 | 0.9502 |
| 0.25 | 0.9479 | 0.9179 | 0.9608 | 0.9608 | 0.9752 | 0.9532 | 0.9624 | 0.9624 | 0.9546 | 0.9546 |
| 0.40 | 0.9540 | 0.9011 | 0.9768 | 0.9588 | **0.9836** | 0.9613 | 0.9733 | **0.9633** | 0.9657 | 0.9607 |
| 0.60 | 0.9477 | 0.9077 | 0.9699 | 0.9299 | 0.9811 | 0.9321 | 0.9681 | 0.9381 | 0.9610 | 0.9410 |
| 0.80 | 0.9345 | 0.8845 | 0.9474 | 0.9074 | 0.9661 | 0.9161 | 0.9580 | 0.9180 | 0.9563 | 0.9363 |

with the basic clustering model in this paper, the results of ACC and NMI are respectively improved by margins of $4.79\%$ and $5.82\%$. Moreover, the ablation study of ULHE-DC suggests that these two types of representation regularization are complementary to each other, and better performance as shown in the last row of Table 2 can be yielded by combining them.

## 4.5 HYPERPARAMETER ANALYSIS

An orthogonal investigation on hyperparameter ($\lambda_1$ and $\lambda_2$) sensitivity is also conducted on MNIST-*full*. Due to the limit of computing resource and time consumption, either of $\lambda_1$ and $\lambda_2$ is set to 5 values, which are around the corresponding empirical best values and results of the 25 experiments are shown in Table 3, in which the above table and the other one respectively represents the results of ACC and NMI from different settings of $\lambda_1$ and $\lambda_2$, i.e. $\lambda_1 \in \{0.02, 0.05, 0.08, 0.13, 0.20\}$ and $\lambda_2 \in \{0.10, 0.25, 0.40, 0.60, 0.80\}$. As seen from Figure 1, $\lambda_1$ is more sensitive than $\lambda_2$ on both ACC and NMI, and it is not appropriate to set $\lambda_1$ with a relatively large value. In brief, it intuitively demonstrates that ULHE-DC maintains acceptable results and relative stability with most reasonable and empirical settings.

## CONCLUSION

In this paper, a uniform quasi-low rank embedding based deep clustering method (ULHE-DC) is proposed. To address the problem of low uncorrelation between different clusters, an inter-cluster uniformity regularization is applied to enhance the discriminability and diversity of the representation structures, which is implemented via the minimization of the hyperspherical energy of the centroids. Additionally, ULHE-DC establishes an intra-cluster compactness regularization to embed features of the same cluster in a quasi-low-rank subspace, and simultaneously improve the instability potentially existing in the optimization of the uniformity regularization loss. Furthermore, an efficient mini-batch based optimization strategy is designed for ULHE to yield better clustering performance. The experimental results show that ULHE-DC outperforms those SOTA approaches.

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

## A    FRAMEWORK OF ULHE-DC

The proposed ULHE-DC method, which performs image clustering based on deep learning, includes two stages, i.e. pretraining and clustering. In the pretraining stage, a fully connected AE is pretrained with the normalized loss $\mathcal{L}_{norm}$ and the reconstruction loss $\mathcal{L}_{rec}$ to learn feasible features. In particular, $\mathcal{L}_{norm}$ enforces the data points embedded on a unit hypersphere. For clustering, the task-specific representation is learned towards the optimization of the clustering objective and the weighted sum of inter-cluster uniformity loss $\mathcal{L}_{unif}$ and intra-cluster compactness loss $\mathcal{L}_{cmpt}$, simultaneously updating the assignments and centroids. $\mathcal{L}_{unif}$ encourages the distribution of cluster centroids as uniform as possible, while $\mathcal{L}_{cmpt}$ is designed to improve the intra-cluster compactness. The framework is shown as Figure 2.

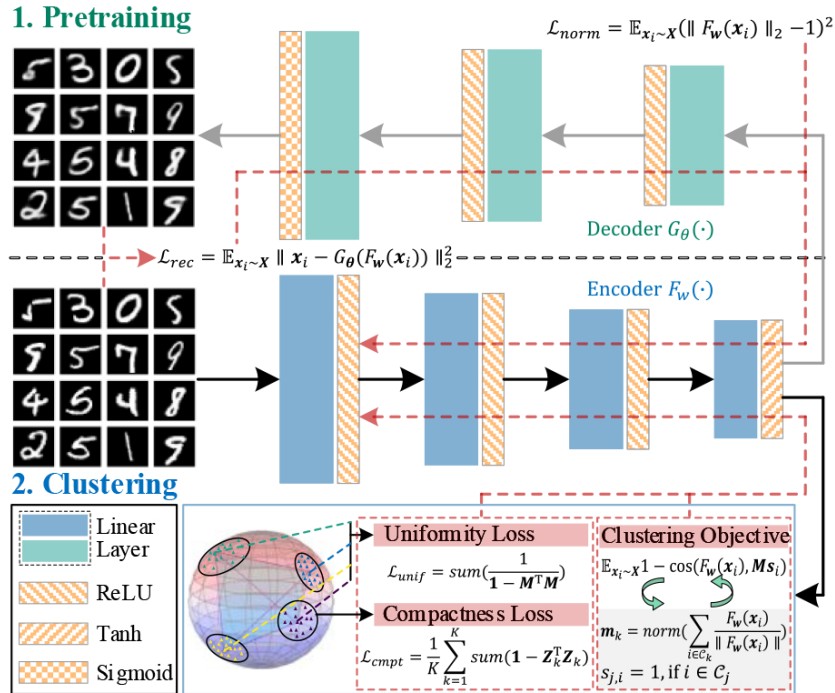

Figure 2: ULHE-DC Framework.

## B    THEORETICAL AND APPLIED ANALYSIS OF THE EXPONENT $v$ IN MHE

In *Sec. 3.3*, the exponent $v$ of the hyperspherical energy has been set to 2. Herein, two aspects of explanations explanation are given.

Firstly, $v$ is not suitable to be set with a large value in theory. Given $K$ cluster centroids, i.e. $M = [m_1, m_2, ..., m_K]^\mathsf{T}$, the hyperspherical energy $\mathcal{E}_v(m_k|_{k=1}^K)$ can be written as follows,

$$\mathcal{E}_v(m_k|_{k=1}^K) := \sum_{i=1}^{K} \sum_{j=1}^{K} f_v(\|m_i - m_j\|)$$

$$= \begin{cases} \sum_{i>j} \|m_i - m_j\|^{-v}, & v > 0 \\ \sum_{i>j} log(\|m_i - m_j\|^{-1}), & v = 0 \end{cases}.$$

**Definition 1** The neighborhood set of the $k$-th centroid $m_k$, signed as $U(k)$, is composed of the indexes of several centroids. If $k' \in U(k)$, it should satisfy the condition $0 < \|m_{k'} - m_k\| < \epsilon$.

According to **Definition 1**, for any $k \in \{1, 2, \ldots, K\}$ and $v > 0$, it can be obtained that

$$\mathcal{E}_v(\boldsymbol{m}_k|_{k=1}^K) = \sum\nolimits_{i>j} \|\boldsymbol{m}_i - \boldsymbol{m}_j\|^{-v}$$

$$= \sum_{i=1}^K \sum\nolimits_{i>j, j\in U(i)} \|\boldsymbol{m}_i - \boldsymbol{m}_j\|^{-v} + \sum_{i=1}^K \sum\nolimits_{i>j, j\in \overline{U}(i)} \|\boldsymbol{m}_i - \boldsymbol{m}_j\|^{-v},$$

where $\overline{U}(i)$ is the complementary set of $U(i)$, i.e. $U(i) + \overline{U}(i) = \{1, 2, \ldots, K\}$. Hence, if $v$ is set with a large value, then

$$\|\boldsymbol{m}_i - \boldsymbol{m}_j\|_{j\in U(i)}^{-v} > \|\boldsymbol{m}_i - \boldsymbol{m}_j\|_{j\in \overline{U}(i)}^{-v}.$$

Actually, the left and right terms of the above inequation respectively denote the local and the approximately global hyperspherical energy. When $K$ and $v$ are relatively large, the minimization of $\mathcal{E}_v(\boldsymbol{m}_k|_{k=1}^K)$ will tend to make the distribution of cluster centers more locally uniform, rather than globally.

Secondly, in the formulation of $\mathcal{E}_v(\boldsymbol{m}_k|_{k=1}^K)$, the calculation of Euclidean distance between the centroids is necessary. But when $v = 2$, MHE can be derived into a concise and intuitive formulation, free of the complex arithmetic exponent.

## C    LEARNING STRATEGY OF ULHE-DC

The training procedure can be clearly compartmentalized to two stage and the summarization of the whole algorithm is presented in Algorithm. 1.

---

**Algorithm 1** Uniform quasi-Low-rank Hypersphere Embedding based Deep Clustering (ULHE-DC)

---

**Input:** Dataset $\mathcal{X} = \{\boldsymbol{x}_i \in \mathbb{R}^D\}_{i=1}^N$, the number of clusters $K$; Maximum iterations $T$.
**Output:** Clustering assignments $\boldsymbol{s}$.
1:  **Step 1 Pretraining**
2:  Initialize $\boldsymbol{w}$ through minimizing Eq. (3), i.e.,
$$\mathcal{L}_{norm-rec} = \mathbb{E}_{\boldsymbol{x}_i \sim \mathcal{X}} \|\boldsymbol{x}_i - G_\theta(F_{\boldsymbol{w}}(\boldsymbol{x}_i))\|_2^2 + \mathbb{E}_{\boldsymbol{x}_i \sim \mathcal{X}} (\|F_{\boldsymbol{w}}(\boldsymbol{x}_i)\|_2 - 1)^2.$$
3:  Initialize $\boldsymbol{M}$ and $\boldsymbol{s}$ through implementing the variant of $k$-means on the embedding $F_{\boldsymbol{w}}(\boldsymbol{x}_i)$.
4:  **Step 2 Clustering**
5:  Initialize hyperparameters $\lambda_0$, $\lambda_1$ and $\lambda_2$.
6:  **for** $t = 1$ to $T$ **do**
7:      Update the cluster assignments $\boldsymbol{s}$ with Eq. (6), i.e.,
$$s_{j,i} = \begin{cases} 1, if\ j = \underset{k=\{1,2,\ldots,K\}}{argmin}\ 1 - cos(F_{\boldsymbol{w}}(\boldsymbol{x}_i), \boldsymbol{m}_k) \\ 0, otherwise. \end{cases}$$
8:      Update the centroid matrix $\boldsymbol{M}$ with Eq. (8), i.e.,
$$\boldsymbol{m}_k^{(t)} = Norm\left(\boldsymbol{m}_k^{(t-1)} + \tfrac{K|\mathcal{C}_k^{(t)}|}{N}\hat{\boldsymbol{m}}_k^{(t)}\right).$$
9:      Update the network parameters $\boldsymbol{w}$ with Eq. (13), i.e.,
$$\min_{\boldsymbol{w}} \mathbb{E}_{\boldsymbol{x}_i \sim \mathcal{X}}[1 - cos(F_{\boldsymbol{w}}(\boldsymbol{x}_i), \boldsymbol{M}\boldsymbol{s}_i)] + \lambda_0 \mathcal{L}_{norm} + \lambda_1 \mathcal{L}_{unif} + \lambda_2 \mathcal{L}_{cmpt},$$
$$s.t.\boldsymbol{s}_i \in \{0,1\}^K, \mathbf{1}^\mathsf{T}\boldsymbol{s}_i = 1.$$
10:     **if** $1 - sum(\boldsymbol{s}^{(t-1)} \odot \boldsymbol{s}^{(t)})/N < \eta$, **then**
11:         Save the parameters $\boldsymbol{w}$ and stop training.
12:     **end if**
13: **end for**

---

## D    CONSISTENCY OF HYPERSPHERE EMBEDDING AND CLUSTERING

One popular working assumption for deep clustering is that the distribution of each class has relatively low-dimensional intrinsic structures, i.e. the equivalent structures of samples are invariant to

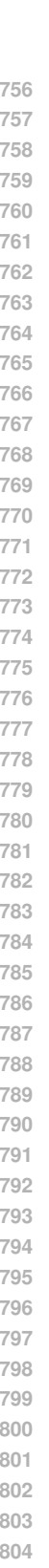

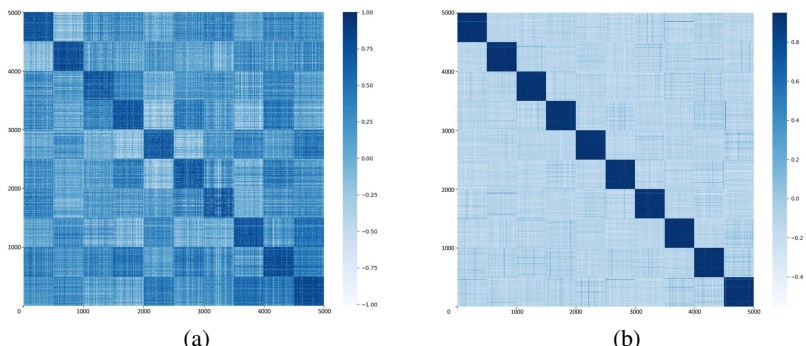

(a)                                         (b)

Figure 3: Cosine similarity between learned features before (**left**) and after (**right**) clustering.

certain classes of deformation. Results shown in Figure 3, Figure 4 (b) and (f), have coarsely supported the assumption. Specifically, we have conducted experiments on 5000 images (500 per class and sorted by class) sampled from MNIST-*test*. Cosine similarity matrices between embeddings before and after clustering were computed and plotted as heatmaps. Though the samples were projected into a hypersphere space, the discriminability of between-class features is relatively obvious in the left of Figure 3, which is consistent with that (the right of Figure 3) after clustering of data.

## E    VISUALIZATION OF ANLATION STUDY

With the t-SNE technique (Laurens & Hinton, 2008), the corresponding clustering visualization on a subset of MNIST-*full* is depicted in Figure 4, including that of the raw data and features extracted by the pretrained $F_w(\cdot)$. It is mainly because that minimizing $\mathcal{L}_{unif}$ could assist the model to learn more discriminative and diverse inter-cluster representations, which can be validated through the observation of Figure 4(c) to Figure 4(f).

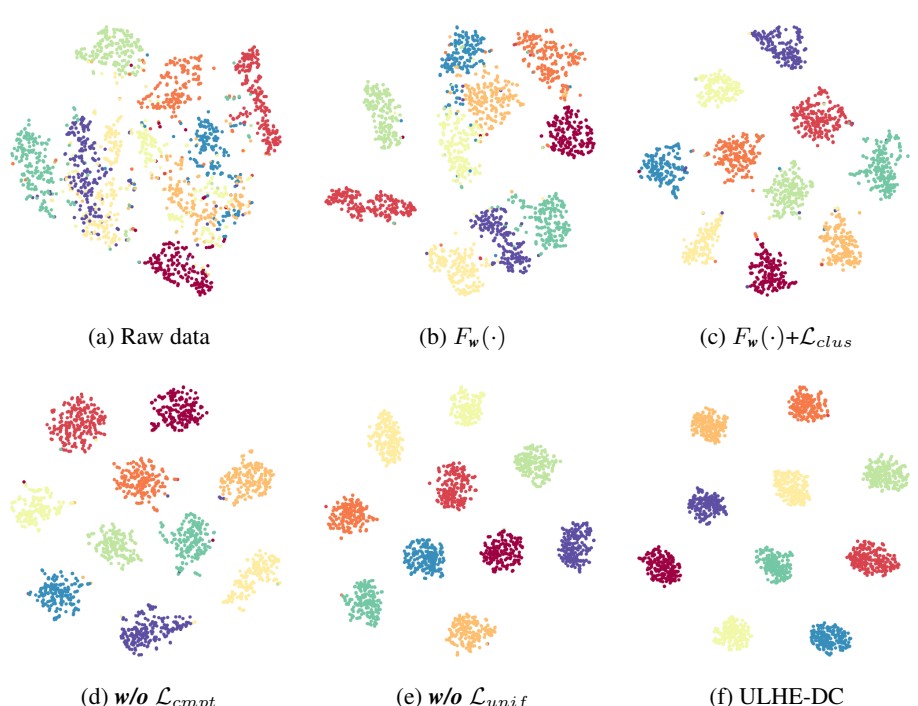

(a) Raw data                  (b) $F_w(\cdot)$                  (c) $F_w(\cdot)+\mathcal{L}_{clus}$

(d) ***w/o*** $\mathcal{L}_{cmpt}$           (e) ***w/o*** $\mathcal{L}_{unif}$           (f) ULHE-DC

Figure 4: Visualization on a subset of MNIST-*full* with 2,000 examples for models in the ablation study.

