# OpenReview forum: "Deep Clustering with Uniform Quasi-low-rank Hypersphere Embedding"
_ICLR.cc/2025/Conference — Submitted to ICLR 2025_

### Official Review · Reviewer_ZCRg · 2024-10-27

**Soundness:** 2
**Presentation:** 2
**Contribution:** 2
**Rating:** 3
**Confidence:** 5

**Summary:**

The paper presents a deep clustering method called Uniform quasi-Low-rank Hypersphere Embedding-based DC. It addresses the insufficient focus on inter-cluster representation structure and the low correlation between clusters, which can degrade clustering performance.  The proposed approach evenly distributed clusters on a unit hypersphere by minimizing the hyperspherical energy, for enhancing the separation between clusters. Simultaneously, the embeddings within each cluster are collapsed into a quasi-low-rank subspace by maximizing intra-cluster correlations, for improving the compactness and similarity of samples within the same cluster. Overall, the paper provides some insights and introduces an interesting perspective to deep clustering methods. Experimental results demonstrate that the performance of the proposed method is promising.

**Strengths:**

1. The proposed method bridges deep clustering with the learning of diverse and discriminative representations. This approach strengthens the notion that enhancing representation learning will be a key direction for the future development of deep clustering methods. Thus, the paper introduces some novel elements to the field.

2. The method shows promising performance on the MNIST datasets, outperforming other competing approaches.

**Weaknesses:**

1. Clarity Issues: The authors should significantly enhance the clarity of their paper. Specific points requiring attention and questions for further clarification are outlined below:
a. The regularization techniques used are based on the Thomson problem, but the paper does not provide background or brief introduction to this concept. Since this is not common knowledge among readers, the authors should offer more context, especially given its central role in the proposed method.
b. In the proof of intra-cluster compactness regularization, the assumption \(r = 1\) is unclear. The authors should provide intuitive reasoning behind this assumption to help readers follow the logic.
c. The norm function used in Equation (7) is unspecified. Given the variety of normalization techniques (e.g., \(L_1\)-norm, \(L_2\)-norm), it is important for the authors to clarify which norm is used to normalize the vector.
d. The paper applies \(L_2\)-norm regularization on the embeddings during pretraining. However, the reasoning behind this choice is not explained. The authors should justify why this regularization is necessary and how it impacts the model’s performance.
e. In the computational complexity analysis, the symbol T is used without explanation. Although \(T_1\) and \(T_2\) are defined, the role of T is not clear and should be explicitly described to avoid confusion.
f. The paper's discussion of the result difference between ACC  and NMI is hard to follow (Line 411-415). The authors should rephrase and simplify this section to improve readability and ensure clarity.
g. I doubt whether this explanation sufficiently justifies the use of uniformity regularization: "it is intuitive that all the clusters are expected to be uniformly distributed in the representation space". This argument is unconvincing to me, as it is not clear why all clusters should necessarily be placed uniformly.

2. Confusing terminology and hard to follow logic flow:
a. The use of "low uncorrelation" in the abstract and introduction is confusing and ambiguous. Typically, we refer to low correlation to indicate that two variables are weakly related. The authors need to clarify the intended meaning of "low uncorrelation" or replace it with more standard terminology.
b.  The logic flow leading to Equation (11) is difficult to follow. The authors should rephrase the derivation to make the steps clearer and easier to understand for readers.

 3. While the paper introduces some novel elements, the overall contribution is relatively modest. The idea of maximizing inter-cluster discriminability and minimizing intra-cluster compactness is very natural in clustering. This work applies that idea using two loss functions—one promoting inter-cluster uniformity and the other enhancing intra-cluster compactness. These losses are inspired by well-known regularization techniques, making the method a thoughtful but familiar extension of existing concepts.

4. The proposed method involves too many hyparameters:
a. The proposed method involves multiple hyperparameters (e.g., \(\lambda_0\), \(\lambda_1\), \(\lambda_2\) in the loss function and the stopping threshold \(\eta\)), which adds complexity. The authors should justify the selection of these hyperparameters more clearly.
b. The ablation studies are incomplete. While the effects of \(\lambda_1\) and \(\lambda_2\) are analyzed, the role of \(\lambda_0\) is not explored. Additionally, the authors should specify the maximum epoch used in the clustering stage, as this is crucial for understanding the computational complexity of the method.

5. Marginal improvement and insufficient experiments:
a. While the method shows significant improvements on MNIST, the gains on USPS and Fashion-MNIST are marginal. The results suggest that the method may not generalize well across datasets.
b. The experimental setup is not comprehensive compared to previous deep clustering studies. Many prior works include small-scale datasets such as FRGC, YTF, CMU-PIE, and COIL, or larger datasets like CIFAR-10 and CIFAR-100. To make the evaluation more robust, the authors do not need to include all these suggested datasets but should consider testing their method on a broader range of datasets, especially larger-scale datasets, to better assess its generalizability.

Generally, the paper presents a clustering method with some interesting elements, but it faces issues in clarity, hyperparameters, and experimental studies. While the approach offers an incremental contribution to the field, the novelty is somewhat limited, and further justification and experimentation are needed to strengthen the work.

**Questions:**

The specific questions and suggestions are outlined in the "Weaknesses" section.

---

### Official Review · Reviewer_nrBp · 2024-10-31

**Soundness:** 2
**Presentation:** 2
**Contribution:** 2
**Rating:** 3
**Confidence:** 5

**Summary:**

The work presented a deep clustering method that explicitly maximizes the discriminability and diversity between different clusters and maximizes the compactness within each cluster.

**Strengths:**

1. The motivation of the proposed method is clearly explained to some extent.
2. The experiments showed that the proposed method has higher clustering accuracy than a few baselines.

**Weaknesses:**

1. A few new terminologies have been clearly defined or explained.
   * For instance, the "quasi-low-rank" is not clear to me. Is it "approximately low-rank"?
   * It seems that the "compactness" considered in this paper is not based on Euclidean distance. Instead, it is related to cosine similarity.
   * In (9), the definition of hyperspherical energy is quite confusing. What is the role of $v$? In line 218, it was stated that $v$ is set to 2, which means the second formula in (9) is never used. As $f_v(\cdot)$ is an energy function, is any example provided?
2. The following important claim is wrong: "most existing DC approaches mainly focus on the first issue and learn suitable embeddings with the DNNs trained through a clustering-oriented loss function". Actually, there have been a few papers focusing on maximizing the inter-class distances, but the authors failed to discuss these works. See the example [1].  I don't think the proposed method has substantial differences regarding the key idea.
3. The $L_{norm}$ regularizer given by (2) does not ensure that $\Vert F_w(x _i)\Vert _2=1$, therefore the claim in Line 266 may not hold. The ablation study did not show the impact of $L _{norm}$. I think the regularizer can be removed if using $F_w(x _i)\/\Vert F_w(x _i)\Vert _2$ in subsequent computations, just like (7).
4. It seems that the authors tried to use low-rankness to explain the role of $L{cmpt}$ given by (12). However, the explanation is not clear or not convincing. The minimum of $L{cmpt}$ can be obtained when all vectors in $Z_k$ are the same. As the $\ell_2$-norm of each vector in $Z_k$ is approximately 1, this loss is essentially the sum of pair-wise distances. A true low-rank regularizer should be something like $||Z_k||_\ast$, i.e., the nuclear norm of $Z_k$.
5. The computational complexity of the proposed method is high (quadratic) due (12). Therefore, it may be time-consuming on large-scale datasets. It is necessary to compare the time cost with baselines.
6. The performance of the proposed method is not SOTA. For instance, the clustering performance on Fashion-MNIST is lower than the method proposed in [1]. Actually, there are more competitors with high clustering performance, which are however not included in the experiments of the current paper.



[1] Cai et al. Unsupervised Deep Discriminant Analysis Based Clustering. 2022.

**Questions:**

Please see the comments about weakness.

---

### Official Review · Reviewer_o2qW · 2024-11-02

**Soundness:** 2
**Presentation:** 2
**Contribution:** 2
**Rating:** 5
**Confidence:** 4

**Summary:**

The paper presents a deep clustering approach, called Uniform quasi-Low rank Hypersphere Embedding based Deep Clustering (ULHE-DC), which is supposed to learn inter-cluster uniform and intra-cluster compact representation within an autoencoder (AE) based pre-train framework. Specifically, the uniformity on hypersphere is learned by minimizing a hypersphere energe of the centroids and the quasi-low-rank subspace is promoted by maximizing intra-cluster correlation. Experiments are conducted on MNIST-full/test, USPS, and FashionMnist, showing improved performance compared to the listed baseline methods.

**Strengths:**

+ It is interesting to learn uniform quasi-low-rank hypersphere embedding for deep clustering, which encourages the centriods to be uniformly distributed on hypersphere and the same cluster squashed in a quasi-low-rank subspace.

**Weaknesses:**

1. Though it sounds very interesting to learn the centriods to be uniformly distributed on hypersphere and to enforce the same cluster squashed in a quasi-low-rank subspace, but at the end of the day, it employs a variant $k$-means scheme to learn the centriods on hypersphere, and employs a so-called energy function to make the centroids as uniform as possible. The  energy function based loss in Eq. (10) might encourage the centroids to be distributed as uniform as possible, but it is not clear how to lead a "quasi-low-rank subspace".

Note that the formulation in Eq. (13) is also problematic. On one hand, it is nothing to do with $s_i$ thus the contraints are redendunt. On the other hand, the optimal solution to problem in Eq.(13) is all the embeddings in each cluster will collapse into a singleton, rather than a "quasi-low-rank subspace".

2. The presentation is not good. Just name a few.
- pp.1: L41: ..."and generally come under the performance degeneration and high computational complexity"
- L230-L240: It is misleading to introduce the "sum" because the dimensions in the formulation is incompatible. In particular, the formulation in Eq. (11) was misleading. Maybe it looks encouraging the centroids to be orthogonal.  So for Eq. (12).
- The reviewer was confusing why a simple normalization step was formulated as a so-called normalized loss?

3. The literature review is not sufficient. For example, OLE and MCR2 is mentioned. But both of them are for deep classification, not for deep clustering. There are some work built on MCR2, e.g., MLC (Ding et al. ICCV'23), and others, but none of them was referred to. Also the contrastive learning based deep clustering methods , e.g., CC (Li et al. AAAI'21), GCC (Zhong et al. CVPR'21), NNM (Dang et al. CVPR'21) are totally missing.

4. Experiments are insufficient. There are four terms in the overall loss function. But the ablation study was considering merely two terms, not mentioning of different value of $v$, $\lambda_0$ and etc.

5. Computation complexity is $O(N^2)$. Thus, it is not able to handle deep clustering task of large dataset. Thus, experiments on more challenging dataset, e.g., CIFAR100, ImageNet, are not given.

**Questions:**

1. It is not clear how to lead a "quasi-low-rank subspace".

2. The formulation in Eq. (13) is also problematic. The optimal solution to problem in Eq.(13) is all the embeddings in each cluster will collapse into a singleton, rather than a "quasi-low-rank subspace". Isn't it the case?

3. The dimensions in the formulation is incompatible. The formulation in Eq. (11) was misleading. So for Eq. (12).

4. The reviewer was confusing why a simple normalization step was formulated as a so-called normalized loss?

5. What about the performance compared to the methods in the missing literature?

6. What are the inflence of the pre-training, the first term, the second term, different value of $v$, $\lambda_0$ and etc?

7. Since that the computation complexity is $O(N^2)$, what about the computation time cost in the listed datasets? Is it able to handle larger dataset, e.g., CIFAR100, ImageNet?

---

### Official Review · Reviewer_XRGw · 2024-11-03

**Soundness:** 2
**Presentation:** 2
**Contribution:** 2
**Rating:** 3
**Confidence:** 3

**Summary:**

This paper focuses on the representation structure between clusters in deep clustering. The author proposes a uniform quasi-low-rank hypersphere embedding-based DC (ULHE-DC) method to solve the performance degradation caused by the low uncorrelation between different clusters. The proposed algorithm is more accurate than the existing SOTA benchmarks.

**Strengths:**

(1) This paper considers the problem of deep clustering from the perspective of promoting uniform learning between clusters and compact learning within clusters, and its innovation is novel.

(2) This paper gives a comprehensive theoretical analysis of the proposed ULHE-DC model, which is logical.

**Weaknesses:**

(1) This paper repeatedly mentions the impact of hard samples on clustering performance at the cluster boundary. Can the author define what level of samples are considered hard samples? In addition, can the impact of ULHE-DC on hard samples in the datasets be instantiated?

(2) For the results in Table 1 that cannot be obtained from the original paper, the authors should conduct experiments to supplement them to observe whether the proposed model has advantages more comprehensively.

(3) Although the author claims to have proposed a more powerful model, I can't find the advantages of the proposed model in the performance comparison in Table 1. Except for the relatively good performance on the MNIST-full dataset, the performance improvement on other datasets is very weak. In addition, how is the 98% improvement on the MNIST-test dataset mentioned in line 405 calculated?

(4) Has the author ever considered why the advantages of ULHE-DC vary significantly on different datasets? This unstable performance makes me wonder whether the innovation of this paper is reliable.

(5) The author only conducted the ablation study and hyperparameter analysis on the MNIST-full dataset. Is the performance of ULHE-DC on this dataset applicable to other datasets?

**Questions:**

See weaknesses above.

---

### Meta-Review · Area_Chair_iZY9 · 2024-12-19

**Metareview:**

This paper proposes a deep clustering method based on uniform quasi-low-rank hypersphere embedding, aiming at learning between-cluster uniformity and within-cluster compactness. Although the proposed method demonstrated effectiveness on classic clustering datasets, all four reviewers expressed concerns regarding the manuscript's clarity, algorithmic novelty, and experiment sufficiency. Since no author rebuttal is provided, I decided to reject this paper.

**Additional Comments On Reviewer Discussion:**

No author response is provided in the rebuttal period.

---

### Decision · Program_Chairs · 2025-01-22

Reject